# Evaluation of the Histone Deacetylase 2 (HDAC-2) Expression in Human Breast Cancer

**DOI:** 10.3390/cancers16010209

**Published:** 2024-01-01

**Authors:** Christos Damaskos, Iason Psilopatis, Anna Garmpi, Dimitrios Dimitroulis, Konstantinos Nikolettos, Kleio Vrettou, Panagiotis Sarantis, Evangelos Koustas, Gregory Kouraklis, Efstathios A. Antoniou, Michail V. Karamouzis, Nikolaos Nikolettos, Panagiotis Tsikouras, Georgios Marinos, Emmanouil Kontomanolis, Konstantinos Kontzoglou, Nikolaos Garmpis

**Affiliations:** 1Renal Transplantation Unit, Laiko General Hospital, 11527 Athens, Greece; 2N.S. Christeas Laboratory of Experimental Surgery and Surgical Research, Medical School, National and Kapodistrian University of Athens, 11527 Athens, Greecenikosg22@hotmail.com (N.G.); 3Department of Obstetrics and Gynecology, University Erlangen, Universitaetsstrasse 21–23, 91054 Erlangen, Germany; 4First Department of Propedeutic Internal Medicine, Laiko General Hospital, Medical School, National and Kapodistrian University of Athens, 11527 Athens, Greece; annagar@windowslive.com; 5Second Department of Propedeutic Surgery, Laiko General Hospital, Medical School, National and Kapodistrian University of Athens, 11527 Athens, Greece; 6Department of of Obstetrics and Gynecology, Democritus University of Thrace, 68100 Alexandroupolis, Greecennikolet@med.duth.gr (N.N.);; 7Department of Cytopathology, Sismanogleio General Hospital, 15126 Athens, Greece; 8Molecular Oncology Unit, Department of Biological Chemistry, Medical School, National and Kapodistrian University of Athens, 11527 Athens, Greece; psarantis@med.uoa.gr (P.S.); vang.koustas@gmail.com (E.K.);; 9Department of Surgery, Evgenideio Hospital, Medical School, National and Kapodistrian University of Athens, 11527 Athens, Greece; 10Department of General Surgery and HPB Surgery of Adults and Children, Hygeia Hospital, 15123 Athens, Greece; 11Department of Hygiene, Epidemiology and Medical Statistics, National and Kapodistrian University of Athens, 11527 Athens, Greece; marinosgiorgos@hotmail.com

**Keywords:** triple, negative, breast, cancer, clinicopathological, histone, deacetylase, HDAC

## Abstract

**Simple Summary:**

Histone deacetylase inhibitors (HDACIs) represent a relatively new drug class with potent regulatory effects on the epigenetics in cancer, ranging from apoptosis induction and cancer cell death to cell cycle arrest. In spite of the fact that HDACIs have, so far, received approval for the treatment of mainly hematologic malignancies, there are numerous preclinical, as well as, clinical trials in the setting of (triple negative) breast cancer with very promising results. In this study, we aimed at assessing the clinical importance of HDAC-2 in triple negative breast cancer. A total of 138 breast cancer specimens were examined immunohistochemically. This current study demonstrated that increased HDAC-2 expression correlates with significant clinicopathological parameters of triple negative breast cancer patients, such as survival, recurrence, and disease stage.

**Abstract:**

Background/Aim: Triple negative breast cancer belongs to the most aggressive breast cancer forms. Histone deacetylases (HDACs) constitute a class of enzymes that exhibit a significant role in breast cancer genesis and progression. In this study, we aimed at assessing the clinical importance of HDAC-2 in triple negative breast cancer. Materials and Methods: A total of 138 breast cancer specimens were examined on an immunohistochemical basis. A statistical analysis was performed in order to examine the association between HDAC-2 and the survival and clinicopathological features of the patients. Results: Increased HDAC-2 expression was observed in every fourth case of triple negative breast cancer with positive HDAC-2 staining, whereas only 12 out of 98 non-triple negative breast cancer samples showed high HDAC-2 expression. HDAC-2 overexpression correlated with prolonged overall survival (OS) and disease-free survival (DFS) in triple negative breast cancer. Conclusions: High HDAC-2 levels in triple negative breast cancer seem to positively influence patient survival, disease stage and recurrence.

## 1. Introduction

In the United States, breast cancer is the leading malignancy and the second most prevalent cancer death cause in female patients [1]. Triple negative breast cancer composes a breast cancer type whose cells do not express receptors for the human epidermal growth factor receptor 2 (HER2) protein or the hormones progesterone and estrogen, and accounts for roughly 15% of all breast cancer [2]. Signs and symptoms of patients with triple negative breast cancer are similar to other common breast cancer types, yet triple negative breast cancer correlates with higher metastasis and recurrence rates, hence accounting for a more aggressive behavior [3]. Consequently, the diagnosis of triple negative breast cancer commonly occurs in advanced tumor stages, while the 5-year survival rate drops to 12%, in the presence of distant metastases [4]. In initial stages, breast-conserving surgery, followed by postoperative radiotherapy, and complete mastectomy represent first-line therapeutic options for localized triple negative breast cancer [5]. Chemotherapy represents the key systemic therapy form, alongside a main element of combined treatment [6], with currently applied chemotherapeutic regimens ranging from taxanes and platinum compounds, to anthracyclines or antimetabolites [7]. Due to the lack of progesterone receptor (PR), estrogen receptor (ER) and HER2 expression, triple negative breast cancer patients may not profit from endocrine and trastuzumab therapy, but require targeted drugs, antibody-drug conjugates, or immunotherapy, especially in advanced tumor stages [4]. 

The fundamental unit of DNA structural organization that makes it easier for genetic material to be packaged into a denser form that fits inside the eukaryotic nucleus is called a nucleosome. More specifically, a positively charged histone octamer made up of two identical copies of each of the four core histone proteins—H2A, H2B, H3, and H4—is wrapped by a negatively charged DNA strand [8,9,10]. The assembly of basal factors to form the preinitiation transcriptional complex is hampered by this condensed formation with low levels of acetylation on the lysine residues of the aminoterminal tails [11,12]. The lysine residues’ positive charge is neutralized by post-translational acetylation of their NH2-terminal tails, which reduces the histones’ affinity for the negatively charged DNA. As a result, transcription and DNA strand uncoiling are possible [13,14]. Histone acetylases (HATs) and histone deacetylases (HDACs) work against each other to modulate histone acetylation [15]. HDACs catalyze the removal of acetyl groups from the NH2-terminal lysine residues of core nucleosomal histones, which results in transcriptional repression and the silencing of tumor-suppressor genes (Figure 1) [16,17]. Consequently, a multitude of studies have demonstrated the connection between histone acetylation/deacetylation and carcinogenesis [18,19], suggesting that dysregulation of histone acetylation may be a significant factor in the onset and spread of human cancer.

There are 18 identified human HDACs, which are divided into 4 classes according to both their function and structure (Table 1). Bearing in mind that these HDACs can affect human DNA, the scientific interest was shifted towards the study of their inhibitors. As a matter of fact, several HDAC inhibitors (HDACIs) have received approval by the U.S. Food and Drug Administration (FDA) and are being used against hematological malignancies [14]. Additionally, numerous HDACIs seem to play promising therapeutic role against a variety of malignancies including non-small cell lung, hepatocellular colorectal, endometrial, pancreatic, thyroid, breast, cervical cancer, and melanomas [10,14,20].

Specifically, for the HDAC-2, it has been shown to be involved in various cancer entities, ranging from melanoma, lung cancer, medulloblastoma, and hematological malignancies, to colorectal, pancreatic, urothelial, and prostate cancer [21]. Interestingly, numerous study groups recently also explored the role of HDAC-2 in breast cancer and highlighted its oncogenic capacities in different breast cancer types [14,22,23,24,25,26]. Hu et al. recently reported that pan-HDAC inhibitors that target the neural precursor cell-expressed developmentally down-regulated 9 (NEDD9)-focal adhesion kinase (FAK) pathway increase breast cancer metastasis in preclinical models, potentially seriously impeding the therapeutic success of these drugs. It is worth considering that pan-HDAC inhibitors have the potential to change the course of breast cancer by enhancing invasion [27]. In addition, by increasing secretory pathway calcium ATPase 2 (SPCA2) expression, HDAC inhibitors encourage the transition from mesenchymal to epithelial in triple negative breast cancer cell lines, indicating that aberrant epigenetic regulation of SPCA2 is linked to poor prognosis and breast cancer metastasis [28]. Treatment with HDAC inhibitors can also cause pro-apoptotic proteins like BAK and BAX to express. Consequently, they can be used in conjunction with tamoxifen, which primarily induces apoptosis in ER-positive breast cancer cells [28]. 

Nevertheless, patient-derived triple negative breast cancer tumor samples are not included in the majority of these studies; instead, HDAC-2 expression has been examined in breast cancer cell lines or xenograft tumor models. This current study aims to investigate the expression of HDAC-2 by immunohistochemistry in tissue samples from triple negative breast cancer patients and its relationship to the clinicopathological characteristics of the tumor and the prognosis of the patient.

## 2. Materials and Methods

### 2.1. Clinical Material

Between 2008–2018, female patients with a breast tumor diameter < 20 mm were included in this study. All patients solely underwent surgery but did not receive (neo-) adjuvant chemo- or radiotherapy, either due to lack of indications or personal choice. Patients included in the current study were only stage I–III patients with recurrent triple negative breast cancer whose cause of death was directly related to disease recurrence. Stage IV triple negative breast cancer patients, patients with tumor size > 20 mm, pre-operatively treated patients, and patients whose death correlated with irrelevant medical conditions were excluded from the study (Figure 1). 

The seventh edition of the American Joint Committee on Cancer (AJCC) Grouping system and the Tumor, Node, Metastasis (TNM) system were used to assess tumor staging [29]. Disease-free survival (DFS) was defined as the time interval between the initial diagnosis and the recurrence of the disease, whereas overall survival (OS) was defined as the duration from surgery to death. Written informed consent was obtained from each patient involved in this study before their clinical data and biological specimens were evaluated. The National and Kapodistrian University of Athens Medical School gave its approval to the study (Approval ethic code: 1718004914).

### 2.2. Immunohistochemistry

Initially, tissue samples from triple negative breast cancer were fixed with formalin and embedded in paraffin. Using rabbit polyclonal anti-HDAC-2 (H-5, Santa Cruz Biotechnology, Santa Cruz, CA, USA, sc-7899) antibodies, the expression of HDAC-2 was assessed by immunohistochemistry. Following the manufacturer’s instructions, the antigen was recovered by heating slides in 10 mM citrate buffer for 15 min. After combining 0.3% hydrogen peroxide with methanol and letting it sit for 30 min at room temperature in the dark, the endogenous peroxidase activity was eliminated. After that, all sections were incubated with anti-HDAC-2 antibodies (H-54, sc-7899, Santa Cruz Biotechnology) at room temperature for one hour at a dilution of 1:200 in phosphate-buffered saline (PBS—Primary Antibody Diluent, ScyTek Laboratories Inc., Logan, UT, USA). Thereafter, there were two more 10 min room-temperature incubations, one with a biotinylated linking reagent and one with a streptavidin label conjugated with peroxidase. Then, to increase immune peroxidase activity, a 3,3′-diaminobenzidine tetrahydrochloride (DAB) substrate kit (UltraVision Quanto HRP Detection System, Thermo Fisher Scientific, Labvision Corporation, Fremont, CA, USA) was used. Hematoxylin was used to stain the sections. The study employed irrelevant anti-serum or primary antibody omission as negative controls, while tissue sections from pancreatic adenocarcinoma with observably elevated HDAC-2 levels served as positive controls. The tumor’s cell proliferative index was assessed using p53 immunohistochemical expression (−: negative, +: low, ++: middle, +++: high).

### 2.3. Evaluation of Immunohistochemistry

Two separate pathologists measured at least 1000 malignant cells per section using immunohistochemistry; they were not aware of the clinical details. The intensity (0: negative, 1: mild, 2: moderate, 3: strong) and percentage of HDAC-2 positive cells (0: negative staining, 1: less than 10%, 2: equal to or more than 10% and less than 33%, 3: equal to or more than 33% and less than 66%, and 4: equal to or more than 66%) were used to evaluate immunohistochemical staining. The cases were then divided into two groups based on the HDAC-2 immunohistochemistry scores, which were calculated by multiplying these two parameters: 0–6 points indicated low HDAC-2 expression, and 7–12 points indicated high HDAC-2 expression. Ki67 staining was regarded positive in case the percentage of positively stained tumor nuclei exceeded 10%.

### 2.4. Statistical Analysis

Standard deviation (SD) was used to express mean values for quantitative variables. Both absolute and relative frequencies are displayed for qualitative variables. The Mann–Whitney test was used for non-normal distribution, while the student’s *t*-test was calculated for mean value comparisons in the case of a normal distribution. Over the course of the follow-up period, Kaplan–Meier survival estimates for the events were plotted. Survival curves were compared using log-rank tests. The Cox proportional hazard model was employed to identify independent variables related to survival and recurrence. From the outcomes of the Cox-regression analyses, hazards ratios (HR) with 95% confidence intervals (95% CI) were calculated. The statistical software SPSS 22.0 (SPSS Corporation, Chicago, IL, USA) was used for the analyses, with a significance level of *p* < 0.05.

## 3. Results

Triple negative breast cancer tissue samples from forty female patients with an average age of 67.3 years (SD = 10.8 years) and a tumor diameter of less than 20 mm were included in the study. The mean tumor size was 14.3mm (SD = 8.3mm) in diameter, and the resection margin was tumor free (R0 resection) with a mean margin of 0.6 mm (SD = 1.1 mm). In total, 90% of the patients had ductal, 30% Grade 2 and 70% Grade 3 triple negative breast cancer. 20% of the cases were at stage I, 30% at stage II and the remaining 50% at stage III. The mean Ki67 value was 38% (SD = 21.1%). The median follow-up period was 4.7 years. Sample demographics, clinical and pathological characteristics are presented in Table 2. To assess the intensity of immunohistochemical staining of the triple negative breast cancer samples, an additional 98 samples of non-triple negative breast cancer from patients who met the study criteria were employed as well.

Increased HDAC-2 expression was observed in 11 (27.5%) of the 40 cases of triple negative breast cancer that showed positive HDAC-2 staining (Figure 2 and Table 3). Tumor-free tissue sections were negative for HDAC-2 expression.

The 40 triple negative breast cancer tissue specimens showed a higher percentage of HDAC-2 overexpression, compared to the 98 non-triple negative breast cancer tissues that met the study criteria (27.5% vs. 12.2%, *p* = 0.029) (Table 3 and Figure 3).

As stated in the study criteria, all patients experienced disease relapse and breast cancer-associated death. Mean DFS was 17.8 months (SD = 2.7 months), with the median being 9 months. Mean OS was 29.9 months (SD = 2.99 months), with the median being 30 months. Kaplan–Meier estimates for both OS and DFS significantly differed for low versus high HDAC-2 expression (log-rank test, *p* < 0.05) (Figure 4).

Multiple Cox-regression analysis for survival showed an 81% lower hazard for patients with high HDAC-2 expression (*p* = 0.014) after adjusting for age, immunohistochemical expression, stage, grade, and Ki67 (Table 4).

Multiple Cox-regression analysis for relapse revealed that patients with high HDAC-2 expression had a 74% lower hazard (*p* = 0.017) and that stage increase by 1 increases the relapse hazard by 4.27 times (*p* = 0.006), after adjustment for age, immunohistochemical expression, stage, grade, and Ki67 (Table 5).

## 4. Discussion

HDAC-2 has been stated to play an important role in various breast cancer subtypes [14,22,23,24,25,26]. In this present study, we investigated the clinical importance of the expression of HDAC-2 in triple negative breast cancer. Our results indicated that triple negative breast cancer shows a higher percentage of HDAC-2 overexpression, in comparison with non-triple negative breast cancer tissues. DFS and OS were positively associated with the overexpression of HDAC-2, with patients with high HDAC-2 expression levels exhibiting significantly lower hazard for survival and relapse. Notably, stage increase significantly increases the relapse hazard.

So far, only three studies have explicitly investigated the role of HDAC-2 in triple negative breast cancer. Xu et al. investigated the expression of HDAC-2 in triple negative breast cancer cells, and found it to be significantly lower in non-triple negative breast cancer than in triple negative breast cancer. However, the survival analysis from the online website GEPIA2 revealed an association of high HDAC-2 expression with poor prognosis in breast cancer patients [22]. Furthermore, Xie et al. found levels of pyruvate dehydrogenase kinase 1 (PDK1), which are upregulated by HDAC-2, to be significantly higher in MDA-MB-231 triple negative breast cancer cells. Of note, a correlation analysis between PDK1 and HDAC levels in breast cancer samples from the Cancer Genome Atlas (TCGA) dataset highlighted that high HDAC-2 expression had a poor OS probability [23]. On the contrary, Garmpis et al. indicated ER- and HER2-positive breast cancer specimens to exhibit higher HDAC-2 levels than PR-positive or triple negative breast cancer specimens, whereas DFS and OS showed a negative association with the overexpression of HDAC-2 [14].

Our study results are, hence, partly contradictory to the outcomes of the aforementioned relevant studies. Nevertheless, these discrepancies may be, first of all, justified by the differences in the chosen study materials. In our study, we incorporated 40 triple negative breast cancer tissue samples from 40 female patients with a tumor diameter of less than 20 mm and compared them with 98 non-triple negative breast cancer tissue samples, whereas the other study groups either used triple negative breast cancer cell lines or extensive breast cancer datasets. In comparison with the previous relevant work of our study group [14], double so many patients with triple negative breast cancer were included in the present study, hence providing a more extensive and consequently more reliable database in terms of statistical analysis. Secondly, the inclusion criteria differed for each study, while the criteria for HDAC-2 expression evaluation were not the same, respectively. Thirdly, the use of different antibodies, as well as different methods in terms of tissue sample processing, might have also led to the above contradictions.

## 5. Conclusions

In summary, this present study demonstrated that increased HDAC-2 expression is associated with very important clinicopathological parameters of triple negative breast cancer patients, such as survival, recurrence and disease stage. Given both the rarity and the aggressiveness of this breast cancer subtype, more research incorporating larger datasets needs to be conducted, in order for HDAC-2 to be defined as a novel trustworthy index of aggressiveness, as well as a target in the context of triple negative breast cancer treatment. Future trials should, thus, focus on the role of HDAC-2 in triple negative breast cancer, by explicitly using or, at least, distinguishing triple negative breast cancer samples from other breast cancer subtypes that evidently exhibit distinctive molecular patterns. This is also a precondition in terms of the future development of effective HDAC inhibitors that could suppress the action of HDACs and express efficient anti-tumor activity, hence paving the way for novel therapeutic approaches in (triple negative) breast cancer treatment.

## Data Availability

The data presented in this study are available on request from the corresponding author.

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
