# Peer review of "Evaluation of the Histone Deacetylase 2 (HDAC-2) Expression in Human Breast Cancer"

_cancers, 2024, doi:10.3390/cancers16010209_

Round 1

Reviewer 1 Report

Comments and Suggestions for Authors

The manuscript by Damaskos et al. is an interesting presentation of clinical data regarding HDAC-2 expression in clinical triple negative breast cancer samples related to survival metrics. This report should be of interest to the general cancer research community as further research may lead to future biomarker discovery and development of new therapeutic strategies.

There are a few issues that need clarification/correction. In the discussion the authors note that their data conflicts with previous reports. Their reasonable argument for conflicting results is based on study design and methods. An acceptable argument, however, this argument does not seem to apply to their own previous publication (reference 15 in the manuscript) where patient selection criteria is functionally identical. This reviewer believes that additional clarification could be provided.

The manuscript is in the wrong journal template. Is in J. Pers. Med., should be in Cancers.

There is a graph missing. Diagram 2 caption as well as the manuscript text states disease free survival (DFS) and overall survival (OS) is presented. However, only DFS is presented.

Table issues. Tables III, IV and V have symbol definitions listed below, but no symbols are present in the tables.

Typographical suggestions.

Line 39: ‘preclinical and, mean-while also clinical’ consider ‘preclinical, as well as, clinical’

Line 84: ‘wrapped around by a negatively charged DNA’ consider ‘wrapped by a negatively charged DNA’

Line 100: Sentences shouldn’t begin with a number, consider ‘There are 18 human HDACs identified, which are…”

Line 107: consider a comma after ‘breast’ delete ‘and’ after ‘cervical’, replace semicolon with comma.

Line 131: ‘they only look into HDAC-2 expression in…’ consider ‘HDAC-2 expression has been examined in…’

Comments on the Quality of English Language

English is fine. A few typos are noted in the comments.

Author Response

The manuscript by Damaskos et al. is an interesting presentation of clinical data regarding HDAC-2 expression in clinical triple negative breast cancer samples related to survival metrics. This report should be of interest to the general cancer research community as further research may lead to future biomarker discovery and development of new therapeutic strategies

There are a few issues that need clarification/correction.

In the discussion the authors note that their data conflicts with previous reports. Their reasonable argument for conflicting results is based on study design and methods. An acceptable argument, however, this argument does not seem to apply to their own previous publication (reference 15 in the manuscript) where patient selection criteria is functionally identical. This reviewer believes that additional clarification could be provided.

We have now added following statement in the discussion: ’’In comparison with the previous relevant work of our study group [15], double so many patients with triple negative breast cancer were included in the present study, hence providing a more extensive and consequently more reliable database in terms of statistical analysis.’’  

The manuscript is in the wrong journal template. Is in J. Pers. Med., should be in Cancers.

We have now corrected the journal template.

There is a graph missing. Diagram 2 caption as well as the manuscript text states disease free survival (DFS) and overall survival (OS) is presented. However, only DFS is presented.

We have now added the missing diagram.

Table issues. Tables III, IV and V have symbol definitions listed below, but no symbols are present in the tables.

We have now corrected the table issues.

Typographical suggestions.

Line 39: ‘preclinical and, mean-while also clinical’ consider ‘preclinical, as well as, clinical’

Line 84: ‘wrapped around by a negatively charged DNA’ consider ‘wrapped by a negatively charged DNA’

Line 100: Sentences shouldn’t begin with a number, consider ‘There are 18 human HDACs identified, which are…”

Line 107: consider a comma after ‘breast’ delete ‘and’ after ‘cervical’, replace semicolon with comma.

Line 131: ‘they only look into HDAC-2 expression in…’ consider ‘HDAC-2 expression has been examined in…’

We have now corrected all typos according to the reviewer’s suggestions.

Reviewer 2 Report

Comments and Suggestions for Authors

The authors examined the association between HDAC-2 and the survival and clinicopathological features of patients in order to investigate the clinical importance of HDAC-2 in triple negative breast cancer (TNBC). They found that HDAC-2 overexpression correlated with prolonged overall survival (OS) and disease-free survival (DFS) in TNBC cases. They concluded that high HDAC-2 levels in TNBC seem to positively influence patient survival, disease stage, and recurrence. This study has a certain significance. However, I have provided some comments and follow-up questions below.

1.     This study did not include any treatment factors, and therefore it seems that the prognosis just reflects the tumor biology itself. Do the authors recommend any specific treatment plan based on their findings?

2.     Although adjuvant treatment was not included in this study, the authors should include the treatment provided after recurrence.

3.     I recommend that the authors study the impact of HDAC-2 on prognosis in cases treated with chemotherapy?

4.     The authors should describe the criteria for the categorization of p53 expression.

5.     The authors should state the rational for using a cut-off point of 10% for the Ki-67 index value because the mean Ki-67 value was 38%, and the value might be higher in the TNBC cases.

6.     The number of TNBC cases examined (n=40) was small.

7.     The authors need to indicate whether the study was just for TNBC cases or for all cases in Diagram 2.

8.     The authors should include whether or not they used the TN subtype as a factor in the multivariate cox-regression analysis for survival or include it in the analysis of the TNBC cases.

9.     The authors should include the median follow-up period.

Author Response

The authors examined the association between HDAC-2 and the survival and clinicopathological features of patients in order to investigate the clinical importance of HDAC-2 in triple negative breast cancer (TNBC). They found that HDAC-2 overexpression correlated with prolonged overall survival (OS) and disease-free survival (DFS) in TNBC cases. They concluded that high HDAC-2 levels in TNBC seem to positively influence patient survival, disease stage, and recurrence. This study has a certain significance. However, I have provided some comments and follow-up questions below.

  1. This study did not include any treatment factors, and therefore it seems that the prognosis just reflects the tumor biology itself. Do the authors recommend any specific treatment plan based on their findings?

HDACs seem to play an important role in both diagnosis and treatment of different cancer entities. In the present case, synthetic HDAC2 enhancers could possibly account for a better OS/DFS in TNBC patients.

  1. Although adjuvant treatment was not included in this study, the authors should include the treatment provided after recurrence.

Upon recurrence, patients were excluded from the study and received chemotherapy according to the guidelines provided by the European Society for Medical Oncology.

  1. I recommend that the authors study the impact of HDAC-2 on prognosis in cases treated with chemotherapy?

      Thank you for this useful suggestion. We are currently working on a novel paper examining cases treated with chemotherapy. 

  1. The authors should describe the criteria for the categorization of p53 expression.

      We have now described the relevant criteria.

  1. The authors should state the rational for using a cut-off point of 10% for the Ki-67 index value because the mean Ki-67 value was 38%, and the value might be higher in the TNBC cases.

      The value was chosen in accordance with the cut-off point used in other publications (see discussion) with a view to ensuring for comparable data.

  1. The number of TNBC cases examined (n=40) was small.

      We agree with the reviewer that the number of TNBC cases is rather small, however few works have so far been published on HDAC2 and TNBC, hence rendering 40 cases in this context an acceptable number of cases that at least allows for some preliminary results.

  1. The authors need to indicate whether the study was just for TNBC cases or for all cases in Diagram 2.

      It is for all cases as stated in the paragraph above Diagram 2.

  1. The authors should include whether or not they used the TN subtype as a factor in the multivariate cox-regression analysis for survival or include it in the analysis of the TNBC cases.

As stated in the text, multivariate cox-regression analysis for survival was based on age, immunohistochemical expression, stage, grade, and Ki67.

  1. The authors should include the median follow-up period.

      We have now included the mean follow-up period.

Reviewer 3 Report

Comments and Suggestions for Authors

This study focuses on evaluating the clinical significance of Histone Deacetylase 2 (HDAC-2) in triple-negative breast cancer (TNBC), known for its aggressive nature. A total of 138 breast cancer specimens underwent immunohistochemical analysis, revealing increased HDAC-2 expression in one-fourth of TNBC cases compared to a lower occurrence in non-triple-negative samples. Notably, elevated HDAC-2 levels were associated with improved overall survival (OS) and disease-free survival (DFS) specifically in TNBC patients. The findings suggest a potential positive influence of high HDAC-2 levels on patient survival, disease stage, and recurrence in the context of triple-negative breast cancer is interesting and opens avenues of triple-negative breast cancer.

Minor Comment: Figure 2, the Authors should include adjacent tissue (or healthy tissue sample) in the figure compared to the IHC of TNBC.

Author Response

Thank you for your useful comment. Unfortunately, we do not possess photos of the healthy tissue samples. However, as stated in the text, increased HDAC-2 expression was observed in 11 (27.5%) of the 40 cases of triple negative breast cancer that showed positive HDAC-2 staining (Figure 2 and Table III). Tumor-free tissue sections were negative for HDAC-2 expression (data not shown).

Round 2

Reviewer 2 Report

Comments and Suggestions for Authors

The title "Evaluation of the Histone Deacetylase 2 (HDAC-2) Expression in Human Triple-Negative Breast Cancer" should be changed to "in Human Breast Cancer".

Author Response

We have now changed the title as suggested by the reviewer

Round 3

Reviewer 2 Report

Comments and Suggestions for Authors

The authors had responded to my comments.